# Clinical Utility of 4C Mortality Scores among Japanese COVID-19 Patients: A Multicenter Study

**DOI:** 10.3390/jcm11030821

**Published:** 2022-02-03

**Authors:** Kazuki Ocho, Hideharu Hagiya, Kou Hasegawa, Kouji Fujita, Fumio Otsuka

**Affiliations:** 1Department of General Medicine, Okayama University Graduate School of Medicine, Dentistry and Pharmaceutical Sciences, Okayama 700-8558, Japan; m03009ko@jichi.ac.jp (K.O.); khasegawa@okayama-u.ac.jp (K.H.); fwkp9270@nifty.com (K.F.); fumiotsu@md.okayama-u.ac.jp (F.O.); 2Department of Internal Medicine and Infectious Disease, Tsuyama Chuo Hospital, Okayama 708-0841, Japan

**Keywords:** COVID-19, clinical score, 4C mortality score, mortality, severity, length of hospitalization

## Abstract

Background: We analyzed data from COVID-19 patients in Japan to assess the utility of the 4C mortality score as compared with conventional scorings. Methods: In this multicenter study, COVID-19 patients hospitalized between March 2020 and June 2021, over 16 years old, were recruited. The superiority for correctly predicting mortality and severity by applying the receiver operating characteristic (ROC) curve was compared. A Cox regression model was used to compare the length of hospitalization for each risk group of 4C mortality score. Results: Among 206 patients, 21 patients died. The area under the curve (AUC) (95% confidential interval (CI)) of the ROC curve for mortality and severity, respectively, of 4C mortality scores (0.84 (95% CI 0.76–0.92) and 0.85 (95% CI 0.80–0.91)) were higher than those of qSOFA (0.66 (95% CI 0.53–0.78) and 0.67 (95% CI 0.59–0.75)), SOFA (0.70 (95% CI 0.55–0.84) and 0.81 (95% CI 0.74–0.89)), A-DROP (0.78 (95% CI 0.69–0.88) and 0.81 (95% CI 0.74–0.88)), and CURB-65 (0.82 (95% CI 0.74–0.90) and 0.82 (95% CI 0.76–0.88)). For length of hospitalization among survivors, the intermediate- and high- or very high-risk groups had significantly lower hazard ratios, i.e., 0.48 (95% CI 0.30–0.76)) and 0.23 (95% CI 0.13–0.43) for discharge. Conclusions: The 4C mortality score is better for estimating mortality and severity in COVID-19 Japanese patients than other scoring systems.

## 1. Introduction

The coronavirus disease 2019 (COVID-19) caused by a severe acute respiratory syndrome coronavirus 2 (SARS-CoV-2) has spread globally with unprecedented rapidity [1]. The disease potentially yields severe illnesses, such as acute respiratory distress syndrome (ARDS) and multi-organ dysfunction syndrome, with a high mortality rate in patients with various risk factors [2]. The underlying conditions for developing a fatal disease include age, hypertension, diabetes mellitus, coronary heart disease, chronic obstructive lung disease, and chronic kidney disease [3]. Therapeutic strategies established over time are currently providing evidence-based treatment for COVID-19 patients [4]. Although the focus is on severe and fatal cases, most COVID-19 patients follow an asymptomatic or mild course [5] without necessary admission management and specific treatment. 

Despite the better understanding of its pathophysiology, a prediction scoring of COVID-19 patients for their severity is yet to be established. The clinical utility of 4C, the Coronavirus Clinical Characterization Consortium mortality score, was advocated by the International Severe Acute Respiratory and Emerging Infection Consortium [6]. According to the literature, the 4C mortality score can determine whether COVID-19 patients require hospitalization. However, the validity of the 4C mortality score has not been fully confirmed in Japanese patients. As have been discussed in various medical fields, racial differences may affect the clinical manifestations, severity, and even prognosis of disease, influencing the usefulness of the scoring system. Conventional scoring approaches for respiratory infections, originally used for assessment of patients who need hospitalization, such as the A-DROP (age, dehydration, respiratory failure, disorientation, and blood pressure) system and CURB-65 (confusion, blood urea >7 mmol/L, respiratory rate ≥30 breaths/min, blood pressure, and age) may be applied to COVID-19 patients [7]; however, this has not been sufficiently evaluated as well. 

This study aimed to investigate the utility of the 4C mortality score as a prognostic prediction score for Japanese patient hospitalized due to COVID-19, especially as compared with other scoring approaches.

## 2. Materials and Methods

### 2.1. Study Design and Settings

This study is a multicenter, retrospective observational study performed in Japan between 1 March 2020 and 30 June 2021. The study recruited consecutive patients aged 16 years and older who were diagnosed with COVID-19 by laboratory tests and hospitalized at the Okayama University Hospital (865 beds) and the Tsuyama Chuo Hospital (515 beds). 

### 2.2. Study Protocol

We collected clinical data on the first day of hospital admission, including age, sex, underlying comorbidities (hypertension, diabetes mellitus, chronic kidney disease, chronic obstructive pulmonary disease, and malignancy), and vital signs at hospitalization (body temperature, pulse rates, respiratory rates, and blood pressure). Later, data on use of respiratory support (provided at their worst conditions), length of hospitalization directly related to COVID-19 (defined by Japanese Ministry of Health, Labor, and Welfare)), and outcome (survival or death directly due to COVID-19) of patients were collected on the discharge day, from medical records. We defined death due to COVID-19 as those patients who passed away during the treatment period, while we excluded those patients who died from other reasons after completing the COVID-19 treatment. Comorbidities were defined according to a modified Charlson comorbidity index, such as coronary disease, diabetes, COPD, connective tissue disease, and renal insufficiency [8]. Obesity, defined as a body mass index (BMI) of more than 30 kg/m^2^, was included as a comorbidity because it was associated with the prognosis of COVID-19 [6]. We stratified the patients into four risk groups according to the previous literature [6]: low-risk (0–3) group, intermediate-risk (4–8) group, high-risk (9–14), and very high-risk (≥15) group (numbers in parentheses are scoring points of the 4C mortality score). All patients were treated in accordance with statement of Japanese Association for Infectious Diseases. No patients received a vaccine against COVID-19.

### 2.3. Outcome Measures

The primary outcome of the present study was to evaluate the accuracy of the 4C mortality score on admission as compared with the accuracies of quick SOFA (sepsis-related organ failure assessment, qSOFA)) [9], SOFA [10], A-DROP [11,12], and CURB65 [13], for predicting mortality and severity of COVID-19. We defined severe patients as those who needed intubation and ventilator support, including those who were clinically considered to require artificial respirator management but refused it. The secondary outcome was to calculate and compare the length of hospitalization (days from hospital admission to discharge) of surviving patients in each risk group of the 4C mortality score. In the case of hospital-acquired COVID-19, we regarded the day of diagnosis as the initial day of hospitalization.

### 2.4. Statistical Analysis

Continuous variables are shown as medians and interquartile ranges (IQRs) and assessed using the Mann–Whitney U test. Categorical variables are reported as numbers and percentages assessed using Fisher’s exact test. By applying the receiver operating characteristic curve, we compared the superiority of each scoring approach to predict mortality and severity correctly. The area under the curve (AUC) with 95% confidential interval (CI) was evaluated as follows: a value of less than 0.5 as no predictive ability, from >0.5 to 0.7 as insufficient, from >0.7 to 0.8 as acceptable, from >0.8 to 0.9 as excellent, and >0.9 as outstanding [14]. We also calculated categorical net reclassification improvement (categorical NRI) of the 4C mortality score for mortality and severity as compared with qSOFA, SOFA, and CURB-65. We drew Kaplan–Meier plots for the length of hospitalization of each risk group and compared them using hazard ratios (HRs) calculated by Cox regression (proportional hazard model). We entered only 2 variables (age and sex) into the multivariable Cox regression models, because of low mortality rate. Data of patients hospitalized for more than 30 days were censored at Day 30. Data were analyzed using EZR software, a modified version of R Commander (version 1.54) based on R [15]. The level of significance was set at a *p*-value of <0.05.

### 2.5. Ethical Approval

Ethical approval was obtained from the Institutional Review Board of Okayama University Hospital (no. 2105-004). The requirement for informed consent was waived because the study was a retrospective analysis of routinely collected data that were anonymized; therefore, individual patients could not be identified.

## 3. Results

We collected data from 206 patients during the study period, of which 185 patients survived, and 21 patients died. Among them, 59 patients required intensive care unit (ICU) admission. The overall median (IQR) age of the patients was 67.5 years (48, 77), and 122 patients (59.2%) were men. No patients had undergone vaccination against COVID-19. A univariate analysis showed that patients in the death group were significantly older than those in the surviving group (65 vs. 79 years, *p* < 0.001) (Table 1). Otherwise, there was no statistically significant difference in the underlying comorbidities between the survivors and the dead. However, the mechanical ventilation use was higher in the dead group (15.1% vs. 42.5%, *p* < 0.004) (Appendix A). 

The numbers and proportions of patients classified into low-, intermediate-, high-, and very high-risk groups by the 4C mortality score in the surviving and death groups are summarized in Table 2. All patients in the low- and intermediate-risk groups survived. Meanwhile, 13.3% of the high-risk group and 50% of the very high-risk group died. 

We compared the 4C mortality score with other plausible risk stratification scores for predicting mortality (Figure 1) and severity (Figure 2) due to COVID-19. No patients with 4C mortality scores of 8 or less died, and the cut-off point of the 4C mortality score was calculated at 9, with a sensitivity of 0.53 and specificity of 1.00. AUC (95% CI) of the ROC study for mortality prediction in the 4C mortality score (0.84 (0.76–0.92)) were higher than those of qSOFA (0.66 (0.53–0.78)), SOFA (0.70 (0.55–0.84), A-DROP (0.78 (0.69–0.88)), and CURB-65 (0.82 (0.74–0.90)) (Figure 1), suggesting the superiority of 4C mortality score. The cut-off point of the 4C mortality score for estimating severity was calculated at 11, with a sensitivity of 0.82 and specificity of 0.74. The AUC (95% CI) of the ROC study for severity prediction of the 4C mortality score (0.85 (0.80–0.91)) were also generally higher than that of qSOFA (0.67 (0.59–0.75)), SOFA (0.81 (0.74–0.89), A-DROP (0.81 (0.74–0.88)), and CURB-65 (0.82 (0.76–0.88)) (Figure 2). Additionally, categorical NRI of 4C mortality scores as compared with qSOFA, SOFA, and CURB-65 for mortality were 0.29, 0.24, and 0.28, respectively. Categorical NRI of those for severity were 0.27, 0, and 0.01, respectively. Cut-off values for mortality were 9 (4C mortality score), 1 (qSOFA), 3 (SOFA), and 2 (CURB-65), which are shown in Figure 1. In addition, cut-off values for severity were 11 (4C mortality score), 1 (qSOFA), 2 (SOFA), and 2 (CURB-65), which are shown at Figure 2.

Finally, we compared the length of hospitalization among surviving patients for the low-, intermediate-, and high- or very high-risk groups by the 4C mortality score (Figure 3). The results of Cox regression suggested that patient age (≥65 years, HR 0.83, 95% CI 0.54–1.29) and sex (female, HR 1.17, 95% CI 0.85–1.62) did not influence the length of hospitalization. The intermediate-risk group had a significantly lower HR (HR 0.48, 95% CI 0.30–0.76) for discharge than the low-risk group. Similarly, high- or very high-risk groups had a lower HR (HR 0.23, 95% CI 0.13–0.43) for discharge.

A multivariate Cox regression analysis of factors contributing to the length of hospitalization of patients with COVID-19 was performed. 

## 4. Discussion

The present research suggests that the 4C mortality score is superior to other risk stratification scores for estimating the mortality and severity of COVID-19 in Japanese patients at hospitalization. According to the categorical NRI analysis, the 4C mortality score appeared to be superior to qSOFA, SOFA, and CURB-65 for predicting mortality, and superior to qSOFA for predicting severity. 

The original literature on the 4C mortality score reported that mortality rates of low-, intermediate-, high-, and very high-risk groups were 1.2%, 9.9%, 31.4%, and 61.5%, respectively [6]. As compared to this, mortalities observed in our patients were relatively lower at 0% for the low- and intermediate-risk groups, 13.3% for the high-risk group, and 50% for the very high-risk group. We assume that the difference in patient backgrounds such as ethnicity and number of comorbidities could explain this discrepancy [16]; further investigation on patient data adjustment may be needed. In addition, stratification of patients by the scoring system can differentiate the length of hospital stay after admission. Thus, we consider that the 4C mortality score can be clinically applied for COVID-19 Japanese patients as well, for example, to indicate the need for hospitalization.

The 4C mortality score was first established by British researchers [6]; scoring system validation studies have been performed worldwide. In Italy, the 4C mortality score was compared with other risk stratification scores, such as COVID-GRAM Critical Illness Risk Score (COVID-GRAM), quick COVID-19 Severity Index (qCSI), and National Early Warning Score (NEWS) [17]. The 4C mortality score had the highest AUC (0.80 (95% CI 0.74–0.85), followed by the COVID-GRAM (0.79 (95% CI 0.72–0.84)), NEWS (0.76 (95% CI 0.70–0.82)), and qCSI (0.749 (95% CI 0.69–0.81). In Lithuania, the 4C mortality score was compared with the conventional intentional care unit (ICU) mortality risk scores [18], and was found to be a good predictor of mortality (0.75 (95% CI 0.69–0.81)), equivalent to the Acute Physiology and Chronic Health Evaluation (APACHE) II. In Japan, the RISE UP score, A-DROP, and Rapid Emergency Medicine Score (REMS) were evaluated among 693 COVID-19 patients with pre-existing cardiovascular diseases and risk factors [19]. The 4C mortality score was superior to other risk stratification scores in terms of all-cause mortality (AUC of the 4C mortality score 0.84 (95% CI 0.80–0.88), RISE UP score 0.82 (95% CI 0.78–0.86), A-DROP 0.78 (95% CI 0.73–0.82), and REMS 0.74 (95% CI 0.69–0.78)), and composite endpoint that included mortality and mechanical ventilation (AUC of 4C mortality score 0.78 (95% CI 0.74–0.81), RISE UP score 0.72 (95% CI 0.68–0.76), A-DROP 0.70 (95% CI 0.65–0.74), and REMS 0.69 (95% CI 0.64–0.73)). Our study corroborated the robustness of the 4C mortality score as a prognostic predictor of COVID-19.

The advantage of the 4C mortality score over other scoring systems is that it evaluates a combination of age, sex, number of comorbidities, respiratory rate, consciousness, and laboratory tests (blood urea nitrogen and C-reactive protein). High age and male sex exacerbate COVID-19 [20,21,22]. Underlying diseases such as diabetes mellitus, hypertension, and respiratory disease remarkably influence the prognosis of the patients [23,24]. Other scores such as qSOFA and SOFA assess vital signs and organ dysfunction alone; this could be a reason for the lower power for predicting COVID-19 prognosis. Although A-DROP and CURB-65 weigh patients’ ages, these scores lack data on sex and comorbidities, and thus, fail to be better than the 4C mortality score. 

Our study also suggests that the 4C mortality score can estimate the length of hospitalization for surviving patients; the higher the scores, the more extended the hospital stay. This result indicates that the 4C mortality score can be used for bed management when the number of COVID-19 patients progressively increase, although the literature has not yet highlighted this point. Therefore, we propose adopting the 4C mortality score to effectively predict each patient’s admission duration to efficiently use limited hospital COVID-19 beds.

The strength of the present study is its multicentered nature, focusing on Japanese patients. However, our study has several limitations. First, our data were retrospectively collected, and the sample size was limited; therefore, the ROC curves of the 4C mortality scores does not seem clearly better than SOFA, A-DROP, or CURB-65, and the generalizability of the results should be assessed by larger studies. Second, we did not incorporate other standard scores, including APACHE II and pneumonia severity index, into the comparison. Third, the emergence of genetic variants that could cause severe infections was not considered. Forth, we did not analyze the influence of the treatment that the included patients received. During the study period, treatment strategies and guidelines had gradually changed, and accordingly we provided appropriate therapies to our patients. This therapeutic change could have affected the results. Despite these limitations, the results indicate the clinical usefulness of the 4C mortality score for Japanese patients.

## 5. Conclusions

In summary, we validated the utility of the 4C mortality score for predicting the prognosis of our cohort. Similar to previous studies reported overseas, the 4C mortality score can be used to better estimate mortality and the length of hospital stay as compared with other conventional scoring systems. Furthermore, we highlight that the 4C mortality score can also improve the quality of patient management in the Japanese medical system.

## Figures and Tables

**Figure 1 jcm-11-00821-f001:**
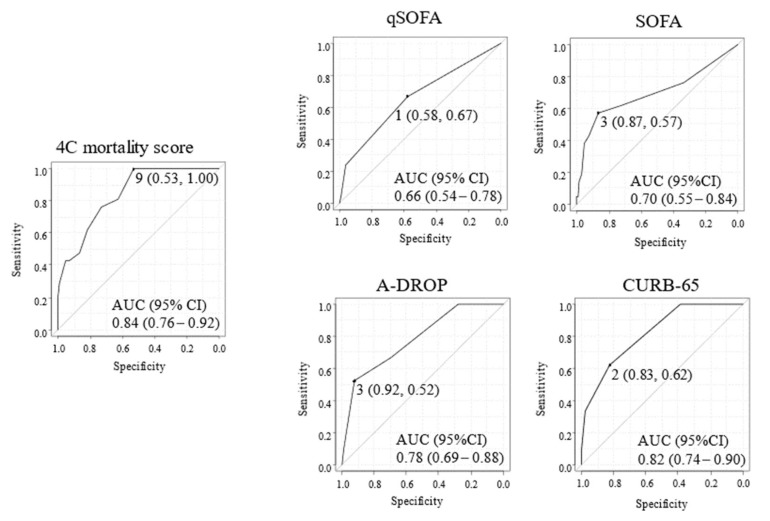
Receiver Operator Characteristic Curve for the mortality of COVID-19 patients. 4C mortality score, the Coronavirus Clinical Characterization Consortium mortality score. qSOFA, A-DROP (age, dehydration, respiratory failure, disorientation, and blood pressure) system. CURB-65 (confusion, blood urea >7 mmol/L, respiratory rate ≥30 breaths/min, blood pressure, and age). qSOFA, quick sepsis-related organ failure assessment. SOFA, sepsis-related organ failure assessment. AUC, the area under the curve. The AUC values for each ROC curve for 4C mortality score, qSOFA, SOFA, A-DROP, and CURB-65, which target is mortality.

**Figure 2 jcm-11-00821-f002:**
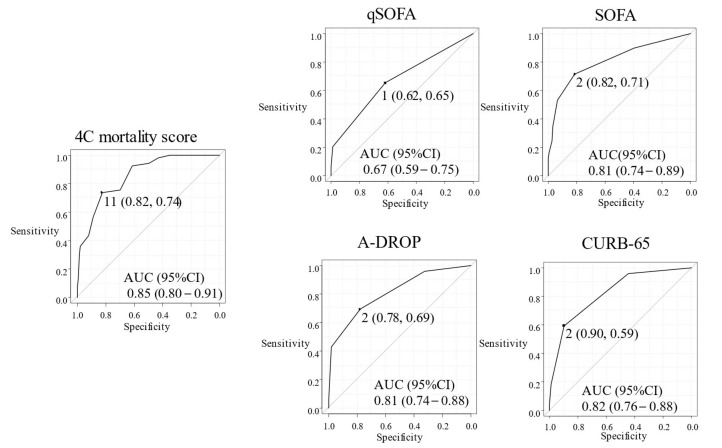
Receiver Operator Characteristic Curve for the severity of COVID-19 patients. The AUC values for each ROC curve for 4C mortality score, qSOFA, SOFA, A-DROP, and CURB-65, which target is severity.

**Figure 3 jcm-11-00821-f003:**
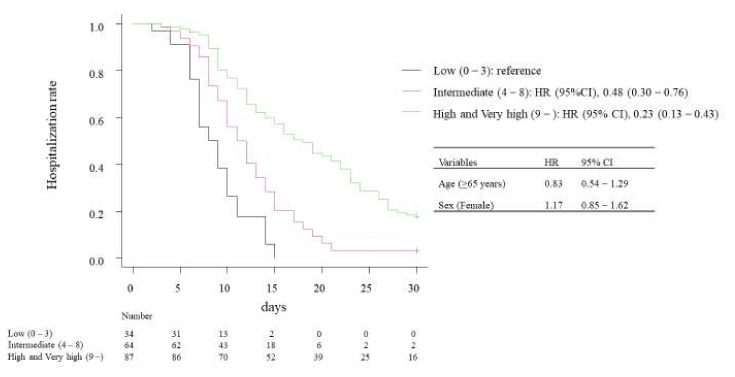
Kaplan-Meier curve of length of hospitalization in survived patients.

**Table 1 jcm-11-00821-t001:** Comparison of the backgrounds of the COVID-19 patients, by outcomes.

	Overall	Survived	Dead	*p*-Value
Number of patients (%)	206	185 (89.8)	21 (10.2)	-
Age, years (IQR)	67.5 (48,77)	65 (46,75)	79 (72,83)	<0.001
Gender (male) (%)	122 (59.2)	110 (59)	12 (57.1)	0.82
Smoking history (%)	81(44)	71 (42.5)	10 (58.8)	0.21
Comorbidities (%)				
Hypertension	114 (55.3)	101 (54.5)	13 (61.9)	0.65
Diabetes mellitus	53 (25.7)	49 (26.5)	4 (19)	0.60
Chronic kidney disease	22 (10.6)	19 (10.2)	3 (14.2)	0.48
COPD	32 (15.5)	27 (14.5)	5 (23.8)	0.34
Malignancy	21 (10.2)	17 (9.2)	4 (19)	0.24

COPD, chronic obstructive pulmonary disease. IQR, interquartile range.

**Table 2 jcm-11-00821-t002:** Numbers of patients classified into the 4C mortality score risk groups on admission.

Risk Group	Total	Survived	Dead
Low (0–3)	34	34 (100%)	0 (0%)
Intermediate (4–8)	64	64 (100%)	0 (0%)
High (9–14)	90	78 (86.7%)	12 (13.3%)
Very high (≥15)	18	9 (50%)	9 (50%)
Overall	206	185	21

## Data Availability

The dataset for this study is available upon request.

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
