# Peer review of "Clinical Utility of 4C Mortality Scores among Japanese COVID-19 Patients: A Multicenter Study"

_jcm, 2022, doi:10.3390/jcm11030821_

Round 1

Reviewer 1 Report

The article discusses the clinical utility of the 4C mortality score in comparison with other scores among Japanese COVID-19 patients in a multicenter study. The article is well-structured. The Abstract and Introduction are clear and relevant for the field. However, some improvements need to be made in the Methodology, Results and Discussion sections. Some observations that the authors can consider are: 

  1. The number of patients included is relatively small, with only 206 patients for a two- center study. Authors should be more specific and use phrases like “in our cohort” instead of “in COVID-19 Japanese patients” to avoid generalization.
  2. The authors included a very heterogeneous population, and it is not clearly stated how many patients were admitted to a COVID-19 clinic and how many required ICU admission.
  3. Of the 206 patients, only 74 required the use of respiratory support. So, the majority of the cases were mild?
  4. The mortality rate is low. From only 21 patients, how many variables did the authors enter into the multivariate models?
  5. A multivariate Cox regression analysis of factors contributing to the hospitalization of patients with COVID-19 was performed. Where are the results exhibited?
  6. The authors state that the 4C mortality score is superior. However, from the ROC curves, 4C does not seem better than SOFA, A-DROP, or CURB-65. How do authors comment on that?

Reviewer 2 Report

Introduction:

  • Authors have to explain the "rationale" of this study. Why it is important to validate the "4C mortality score" in the Japanese population.

Methodology:

  • The authors indicate that the study was conducted between March 1th, 2020 and June 30th, (15 months). According to the authors, the recruitment of patients was consecutive if they met the inclusion criteria. However, authors only included 206 patients. It´s surprising that only 206 patients with severe COVID were admitted in two large Japanese hospitals during COVD-19 pandemic. The authors should confirm these data and explain them correctly, to avoid doubts about "selection bias", since it was a retrospective study and the number of patients included seems low.
  • Author have to explain when the general data of patients were collected, and when the patients were stratified in the different scores: first day of hospital admission or days after. If the time of data collection was different, explain the criteria used.   
  • Regarding the length of hospitalization, authors have to explain whether the duration of hospitalization was only directly related to COVID-19, or if in some cases was related to other comorbidities. If so, how can it influence in the results?
  • Regarding mortality, explain if there were cases of mortality related to patient comorbidities, palliative sedation or any other cause, not related directly with COVID
  • An important point is the influence of COVID therapies on the outcome of patients, taking into account that the study period was 15 months. During this period, some COVID treatments improved the evolution of the patients. In the article, no reference is made to therapies used, only to the use of mechanical ventilation. In addition, it is likely that some of the patients included in 2021 would have received a vaccine against COVID-19 that prevents against serious disease.
  • In summary, previously to the article acceptance, authors have to clarify the  mentioned points, explaining that any of them had an impact on the study conclusions.  If yes, please include in the study limitations section.

Reviewer 3 Report

The paper presented is timely topic and well written. Tha statistical analysis is appropriate and the results clear presented.

here my more specific comment:

-Please determine the net reclassification improvement of the analysed score comparing the q-SOFA, sofa, curb-65.

-The CURB-65 is a score which has been developed for the assessment of patients needing hospitalization for pneumonia

-Please revise English language

Round 2

Reviewer 1 Report

The paper is accetpable for publication 

Author Response

Point #1:

The paper is accetpable for publication 

Response #1:

We would like to thank you for your careful review of the manuscript, constructive comments, and acceptance to publish at Journal of Clinical Medicine